# Evaluation of Blood Lactate, Heart Rate, Blood Pressure, and Shock Index, and Their Association with Prognosis in Calves

**DOI:** 10.3390/vetsci11010045

**Published:** 2024-01-20

**Authors:** Hélène Casalta, Calixte Bayrou, Salem Djebala, Justine Eppe, Linde Gille, Kris Gommeren, Eva Marduel, Arnaud Sartelet, Celine Seys, Jérôme Versyp, Sigrid Grulke

**Affiliations:** 1Clinical Department of Production Animals, Faculty of Veterinary Medicine, University of Liège, 4000 Liège, Belgium; calixte.bayrou@uliege.be (C.B.); justine.eppe@uliege.be (J.E.); asartelet@uliege.be (A.S.); celine.seys@uliege.be (C.S.); jerome.versyp@uliege.be (J.V.); 2Independent Researcher, Muckerstaff Granard, N39AN52 Co Longford, Ireland; salemdjebala@gmail.com; 3Independent Researcher, 1170 Watermael-Boitsfort, Belgium; 4Clinical Department of Companions Animals, Faculty of Veterinary Medicine, University of Liège, 4000 Liège, Belgium; kris.gommeren@uliege.be; 5Independent Researcher, 44200 Nantes, France; eva.marduel@chv-atlantia.com; 6Clinical Department of Equines, Faculty of Veterinary Medicine, University of Liège, 4000 Liège, Belgium; sgrulke@uliege.be

**Keywords:** cattle, hemodynamic parameters, hyperlactatemia, hypotension, oscillometric blood pressure measurement

## Abstract

**Simple Summary:**

Young calves suffering from neonatal diseases received as emergencies often require resuscitation as they are in cardiovascular shock resulting in dehydration or generalized sepsis. The objectives of our study were to evaluate the various tools available to assess shock, and to determine their relationship to survival in calves referred as emergencies. Clinical parameters, a blood analysis, and blood pressure were measured upon arrival at the Veterinary Clinic for Ruminants of Liège on 45 calves from 1 day to 4 months old suffering from various diseases. The results indicate that higher blood lactates upon arrival were associated with an elevated heart rate, increased shock index (defined as the heart rate to systolic blood pressure ratio), and higher clinical shock score. Furthermore, higher initial blood lactate and the shock index were correlated with increased mortality in calves suffering from digestive diseases. In conclusion, the evaluation of specific shock parameters, such as blood lactates and blood pressure in calves could help clinicians to identify critical patients and improve outcomes through better shock management.

**Abstract:**

Diseases in young calves received as emergencies are often associated with hypovolemic and/or septic shocks. The objectives of our study were to (1) assess the correlation between clinical hemodynamic parameters and blood L-lactates (LAC), systolic blood pressure (SBP), and the shock index (SI) recorded upon arrival; and (2) to evaluate how these parameters were related to short-term outcomes in calves under 4 months of age presented as emergencies. We conducted a single-observer prospective observational study on calves aged from 1 day to 4 months, presented to the Veterinary Clinic for Ruminants of Liège from December 2020 to May 2022. Forty-five calves were included in the study. The statistical analysis revealed a significant correlation between LAC and heart rate (r = 0.570; *p* < 0.05) and LAC and SI (r = 0.373; *p* < 0.05). A high LAC value at admission was significantly associated with a negative outcome (death) (*p* < 0.05). In calves suffering from obstructive digestive diseases, the SI was related to the outcome and the analysis indicated a cut-off value of 1.13 (Se = 0.77, Spe = 1). In conclusion, in our study, the initial blood L-lactate value was correlated with heart rate, the shock index, and the clinical shock score, and admission hyperlactatemia was associated with a poor prognosis in calves under 4 months of age. In this cohort, the shock index in calves suffering from digestive diseases was linked with mortality.

## 1. Introduction

The early identification of shock during the initial triage and evaluation of patients presented as emergencies is essential but can be challenging [1,2]. Shock is defined as a severe imbalance between oxygen supply and demand, resulting in inadequate cellular energy production, cellular death, and multiorgan failure [3]. According to Weil’s classification, acute circulatory failure can result from one or a combination of four mechanisms: hypovolemic, cardiogenic, obstructive, and distributive shock [2,4,5,6]. In human emergency medicine, distributive shock (e.g., patients in sepsis or anaphylaxis) is the most frequently encountered, and is often associated with clinically significant vasodilation and inflammation [4]. In ruminants, obstructive and non-obstructive digestive diseases and neonatal septicemia, all of which are frequently associated with hypovolemic or septic shock, are economically important problems encountered when raising young calves [5]. Hypovolemic shock results from the decrease in effective circulating blood volume due to internal or external fluid loss. The subsequent decrease in preload leads to a decreased cardiac output. Septic shock is caused by a drastic drop of systemic vascular resistance and filling pressure following inappropriate vasodilation and increased endothelial permeability, secondary to bacteremia or endotoxemia and the subsequent activation of the inflammatory cascade. This dysregulated host response to infection may also result in cytokine-mediated cardiac and mitochondrial dysfunction, and the combination of both cellular dysfunction and cardiovascular compromise is associated with a significant high mortality risk [2,3,4]. Numerous studies in human and animal intensive care, both in adults and juvenile (pediatric and foal) patients, have demonstrated that early recognition and aggressive resuscitation of shock are associated with better outcomes [2,7,8,9,10,11]. Shock should be suspected based on altered clinical perfusion parameters, which include the level of consciousness, heart rate, mucous membrane color, capillary refill time, temperature, pulse pressure, and urine output [2,4,5]. Clinical predictive models have been developed, especially in equine medicine, with heart rate and abnormal mucous membranes as the most commonly included variables to predict short-term survival outcomes [12]. Although these variables are readily available, they lack sensitivity and specificity [1,4]. Arterial blood pressure (ABP) and lactate are often applied as upstream and downstream markers of tissue perfusion to allow for the early detection of shock. Blood pressure is only an indirect marker of perfusion, and hyperlactatemia may arise in patients with blood pressures within the reference range, although hyperlactatemia was shown to be related to lower ABP [13,14]. Patients presenting both hypotension and hyperlactatemia have higher illness severity scores and lower survival rates [13,15,16]. Hyperlactatemia is typically observed in cases of acute circulatory failure, and numerous studies in different critically ill patient populations have demonstrated the prognostic value of lactate levels [17,18,19,20]. Recent studies performed in dogs and cattle have demonstrated that lactate clearance is superior to a single admission plasma lactate concentration to assess a prognosis [21,22,23,24,25]. The shock index (SI), defined as heart rate (HR) to systolic arterial blood pressure (SBP) ratio, was developed as a simple method to assess shock severity and response to therapy in critically ill people, and was validated in dogs presenting as emergencies [1,22,26,27,28,29,30]. However, the SI has never yet been studied in cattle.

The objectives of our study were to (1) evaluate the correlation between clinical perfusion parameters (heart rate, capillary refill time, pulse), and blood L-lactate concentration, arterial blood pressure, and the shock index (defined as heart rate to systolic blood pressure ratio) recorded at arrival; and (2) to assess their prognostic information in calves under 4 months of age presented as emergencies.

## 2. Materials and Methods

### 2.1. Animal and Clinical Evaluation

All calves, aged from 1 day to 4 months of age, presented to the Veterinary Clinic for Ruminants of the University of Liège (Belgium) from December 2020 to May 2022 were eligible for inclusion. All calves from this range of ages were included, whether they were suspected to be in shock or not.

A clinical examination was performed and the clinical perfusion parameters (heart rate (HR), capillary refill time (CRT), pulse) were recorded. The CRT was subjectively reported as inferior or superior to 2 s, whilst pulse was subjectively defined as normal or weak. 

Calves were divided into 3 categories according to their underlying condition: non-obstructive digestive disease (NOD), obstructive digestive disease (OD), and non-digestive disease (ND). Obstructive digestive diseases were suspected when calves showed abdominal pain, abdominal distension, and scant feces (Figure 1).

### 2.2. Ancillary Tests

One milliliter of jugular blood was sampled in a heparinized syringe and analyzed immediately to obtain the admission venous L-Lactate concentration (LAC) using an EPOC^®^ blood analyzer (Siemens Healthcare, Ottawa, ON, Canada). Hyperlactatemia was defined as LAC > 2 mmol/L.

Systolic, diastolic, and mean arterial blood pressure (respectively, SBP, DBP, and MBP) were obtained using a non-invasive method (oscillometric BP technique, Lightning Multi-Parameter Monitor, Vetronic^®^ Service Ltd., Newton Abbot, UK). The cuff was positioned on the cranial portion of the tail, and a cuff size was chosen with a width approximating 40% of the tail circumference, in accordance with the American College of Veterinary Internal Medicine (ACVIM) guidelines for blood pressure measurement. Hypotension was defined as SBP ≤ 90 mmHg. The shock index (SI) was calculated by dividing the HR by the SBP.

### 2.3. Clinical Shock Evaluation

Clinically suspected shock (CSS) was defined as 2 or more of the following clinical findings: HR > 120 bpm, CRT > 2 s, and weak pulse, with calves being grouped accordingly (clinically suspected shock, CSS) and without clinically suspected shock (WOCSS). 

### 2.4. Therapy and Outcome

Calves were treated at the clinician’s discretion, and complementary tests and medical or surgical treatments were performed accordingly. A positive outcome (survival) was defined as discharge from the hospital. The group of animals with a negative outcome (non-survival) consisted of calves that died or were euthanized during hospitalization because of fatal intraoperative findings, deterioration of general condition, or animal welfare reasons.

### 2.5. Statistical Analysis

The results were presented as mean and standard deviation (SD) and range for the quantitative variables and as numbers for the qualitative variables. In case of non-normality of the distribution (assessed by the Shapiro–Wilk test), the results were reported as median and interquartile range (IQR).

The association between continuous parameters (HR, LAC, SBP, DBP, MBP, SI) was measured by the Spearman correlation coefficient. The Kruskal–Wallis bilateral test was used to compare parameters between groups, and the Chi-squared test was used to compare two qualitative variables. Dunn’s post hoc test was used to analyze the specific sample pair for stochastic dominance.

The admission blood LAC was studied as a continuous variable for correlation with other parameters and with outcomes. Similarly, SBP, DBP, and MBP were studied as continuous variables for correlation with other parameters and outcomes. The outcome was also evaluated depending on the presence or absence of hyperlactatemia (LAC > 2 mmol/L) and hypotension (SAP ≤ 90 mmHg).

A receiver operating characteristic (ROC) curve was used to determine the area under the curve (AUC) for parameters that were significantly related to outcome. The ideal cutoff for sensitivity and specificity was characterized by the Youden index. 

The results were considered significant at the 5% uncertainty level (*p* < 0.05). Statistical analyses were performed using commercially available software (Xlstat, https://www.xlstat.com (accessed on 31 October 2023), Addinsoft Lumivero 2023).

## 3. Results

### 3.1. Demographic Data

Between December 2020 and May 2022, 45 calves were recruited in the study from 45 different herds. The breeds were mainly Belgian blue (41 calves), but two calves were Holstein-Friesian, one Jersey, and one Limousine. The median age and weight were, respectively, 18 days (range 3–113) and 66 kg (range 23–132), with 13 females and 32 males. Twenty-two calves (49%) were referred with obstructive digestive diseases (final diagnosis of abomasal volvulus, caecum volvulus or torsion, mesenteric volvulus, jejunal or coli atresia, intestinal intussusception, strangulated umbilical hernia, peritonitis); 11 calves (24%) were referred with non-obstructive digestive diseases (enteritis); and 12 calves (27%) recruited were suffering from non-digestive diseases like omphalitis without associated peritonitis, urethral rupture, arthritis, malnutrition, and respiratory distress syndrome in newborn calves.

### 3.2. Shock Evaluation

Clinical shock was suspected in twenty-eight calves, hyperlactatemia was present in twenty-eight calves, hypotension in eight calves, and four calves in the study were presenting both.

Tachycardia was highlighted in 28 calves, 31 calves showed CRT > 2 s, 26 had a weak pulse, and 16 calves presented all three parameters (weak pulse (WP), CRT > 2 s, and tachycardia (TC)).

The Spearman correlation test showed a significant linear correlation between lactate concentrations and heart rate (r = 0.570; *p* < 0.05), and the shock index (r = 0.373; *p* < 0.05, respectively) (Figure 2). 

The calves’ clinical and biochemical parameters depending on the clinical shock status (CSS or WOCSS) and the type of disease the calf suffered (NOD, OD, and ND), with *p*-values associated, are reported in Table 1. 

The suspicion of clinical shock (CSS) was associated with a higher LAC (*p* < 0.05) but was not related to SI or ABP. Heart rate and LAC were significantly associated with the type of disease (*p* < 0.05). Heart rate and LAC in calves suffering from OD were higher than in calves suffering from NOD or ND (*p* < 0.05) (Table 1).

### 3.3. Outcome

The clinical and biochemical continuous parameters of all calves involved in the study and their correlation with the outcome were reported in Table 2. 

High blood l-lactate at admission was significantly associated with a negative outcome (*p* < 0.05). An ROC curve analysis demonstrated an AUC = 0.677 with a sensitivity and specificity, respectively, of 0.78 and 0.52 for a LAC cut-off value of 2.05 mmol/L. 

Table 3 presents the proportion of calves in each group regarding the presence or absence of hypotension (SBP ≤ 90 mmHg), hyperlactatemia (LAC > 2), or the combination of hypotension and hyperlactatemia (SBP ≤ 90 mmHg and LAC > 2 mmol/L), and their relationship with the outcome. 

Negative outcomes were significantly more frequent in the group of calves with LAC > 2 mmol/L (*p* < 0.05).

There was a statistical difference in outcomes among calves suffering from NOD, OD, and ND (*p* = 0.027), with a positive outcome being statistically more frequent in calves suffering from NOD than those suffering from OD or ND. 

In the group of calves suffering from digestive diseases (OD and NOD), LAC, HR, and SI were significantly associated with the outcome (*p* < 0.05). A ROC curve analysis for LAC showed an AUC = 0.704 (95% CI 0.523–0.884) with a sensitivity and specificity of 0.93 and 0.41 for a cut-off value of LAC = 1.7. A ROC curve analysis for HR showed an AUC = 0.739 (95% CI 0.567–0.911) with a sensitivity and specificity, respectively, of 0.75 and 0.706 for a cut-off value of HR = 136 bpm. The highest AUC was found for SI (AUC = 0.826, 95% CI 0.672–0.980), with a sensitivity and specificity of, respectively, 0.73 and 0.85 for a cut-off value of SI = 1.18. In calves suffering from an obstructive digestive disease, the SI was related to the outcome. A ROC curve analysis showed AUC = 0.892 (95% CI 0.751–1.000) and a sensitivity and specificity, respectively, of 0.77 and 1 for a cut-off value of SI = 1.13.

## 4. Discussion

Calves displaying modified clinical hemodynamic parameters (HR, CRT, pulse), indicative of shock, were more likely to present elevated LAC levels. Circulatory shock represents a generalized form of acute circulatory failure, resulting in inadequate tissue perfusion and subsequent overproduction of lactates due to oxygen deficiency. Hyperlactatemia defined as LAC > 2 mmol/L is typically present in shock, and a human medicine consensus on circulatory shock recommends blood lactate measurements in all cases of clinically suspected shock [2,4,18,19]. Although other conditions unrelated to shock can lead to hyperlactatemia, such as hepatic failure, seizures, toxins, and medications, these conditions are rarely reported in calves. Specific conditions in calves, such as respiratory diseases, are also frequently associated with hyperlactatemia [31,32,33]. Respiratory distress syndrome in calves is caused by a combination of a lack of surfactant and hypoxia that will lead to respiratory failure [34]. The associated clinical symptoms are polypnea, tachycardia, hypoxemia, hypercapnia, and a mixed metabolic and respiratory acidosis. Hyperlactatemia is frequently associated because of hypoxemia and vasoconstriction associated with hypercapnia, and circulatory shock is frequently associated with respiratory distress syndrome [35,36,37,38].

The elevation of lactates in calves with an elevated heart rate can be explained by the link between tachycardia and shock. Tachycardia may occur in the case of hypovolemia or maldistribution to maintain the cardiac output. Heart rate is a part of clinical hemodynamic parameters suggestive of shock, and its inclusion in the shock index (SI) and clinical shock score (CS) may explain the correlation between LAC and SI and CS. However, using heart rate as a single variable for the diagnosis or management of shock is not recommended due to various causes, such as stress and pain, which can lead to tachycardia (author’s quote).

Our study found no correlation between LAC and arterial blood pressure, aligning with most studies on circulatory shock, where hyperlactatemia indicates an anaerobic metabolism [1,2,4,18,19]. Normolactatemia in hypotensive patients may indicate sufficient organ perfusion to meet the cellular needs despite a low measured peripheral blood pressure [13,39]. A combination of different parameters seems more accurate in evaluating the shock status, and the SI, defined as the HR/SBP ratio, is frequently suggested to assess shock and outcome in human and small animal emergency medicine [21,22,26,27].

Our study showed that blood L-lactates and heart rate were higher in calves with obstructive digestive diseases than in those with non-obstructive digestive diseases or non-digestive diseases. Obstructive digestive diseases often involve ischemic or strangulated lesions, leading to altered intestinal barriers and the potential development of septic and endotoxic shock, in addition to hypovolemic shock from dehydration (lack of fluid intake, fluid loss, or sequestration of fluids in the digestive tract). This supplementary risk could explain the higher LAC and HR values. The pain associated with obstructive digestive diseases could also cause an increase in heart rate and lactate values. Both parameters (HR and LAC) are used in horses with colic as indicators for surgical decision-making and prognosis [40,41,42,43]. 

Finally, our study results indicated that calves with high admission venous L-lactates had a poorer outcome. A similar study about dogs and cats admitted to the emergency room of a teaching hospital demonstrated that hyperlactatemia (LAC > 2 mmol/L) was associated with mortality within 4 h following admission [44]. The association between admission hyperlactatemia and poor outcome has been demonstrated repeatedly in human medicine and in horses and small animals [6,12,18,19,20,45,46]. Recent studies also suggest that admission hyperlactatemia may be related to outcome in adult cows with digestive diseases [25,47,48]. However, the results were not as clear-cut in studies performed on calves. Lausch and colleagues showed that the ability of the preoperative measurement of plasma L-lactate to predict a negative outcome required knowledge of a definite intraoperative diagnosis in calves with acute abdominal emergencies [23]. They also showed that the predictive accuracy of plasma L-lactate was improved in calves under one week of age. In our study, LAC predictive performance was not related to the age of the calves or the diagnosis. However, the sensitivity and specificity of the test with a cut-off value of LAC = 2.05 mmol/L provided by the Youden index were relatively low (sensitivity = 0.78 and specificity = 0.52). Most studies emphasize the importance of a combination of parameters for evaluating shock and outcome. While admission blood lactate values have prognostic significance [17,49], consecutive LAC measurement and lactate clearance in response to therapy typically provide greater insight into patient survival [17,21,22,24,44,45,48,49]. 

In our study, a statistical analysis of contingency tables showed a high positive predictive value with 100% of mortality in calves that were both hyperlactatemic and hypotensive (Table 3). This result is similar to a study performed on cats in the intensive care unit, where hypotensive cats with normolactatemia had higher blood pressure and higher survival than hypotensive cats with hyperlactatemia [13]. This result highlights that despite the absence of the prognostic value of arterial blood pressure alone, the detection of hypotension combined with hyperlactatemia could increase the accuracy of LAC predictive ability. Furthermore, in calves with digestive diseases (OD and NOD), not only LAC but also HR and SI were prognostic factors. Calves with higher SI, HR, or LAC suffering from digestive diseases had a higher risk of a negative outcome. Numerous studies performed on horses suffering from obstructive digestive diseases showed a significant association between heart rate at admission and short-term survival [12,41,43,50,51]. A systematic review about clinical predictive models in horses showed that the most commonly included predictors of short-term survival were heart rate, packed cell volume, and mucous membrane characteristic (color and capillary refill time) [12]. However, none of these studies reported arterial blood pressure as a predictor of short-term outcome. In this group of calves, the SI seemed to have the best predictive value, with an AUC = 0.826 (95% CI 0.672–0.980) and sensitivity and specificity, respectively, of 0.73 and 0.85 for a cut-off value of 1.18. In calves with obstructive digestive diseases, LAC and HR were not related to outcome, but calves with SI > 1.13 had a 100% risk of death (AUC = 0.892, 95% CI 0.751–1.000, sensitivity = 0.77 and specificity = 1 for a cut-off value of 1.13). These results align with the small animal critical care literature, where a higher SI has been associated with survival following trauma in dogs. Similarly, a higher SI was found in dogs in shock with hyperlactatemia or blood loss [1,26,29,30]. In human medicine, the SI is also a useful tool to detect early blood loss, and was associated with outcomes in children presenting with sepsis or septic shock [27,28]. 

## 5. Limitations of the Study

Our study on shock evaluation and prognosis in calves had several limitations. Some clinical parameters, such as mental status or temperature of the extremities, which could have been helpful in evaluating the shock status in calves [2,5], were recorded during clinical examination but not utilized to determine suspected clinical shock in our study. In young calves, the alteration of mental status could also be influenced by an ion imbalance related to diarrhea but not necessarily linked to hypovolemic shock, and this could lead to an incorrect shock assessment. In the recent literature, heart rate, capillary refill time, and pulse pressure were mainly included in the definition of clinical shock scores, but the temperature of extremities was not used to define clinical shock in those studies and did not seem to be associated with survival rate [12,21,22,52]. 

Heart rate and blood L-lactate are parameters related to shock in all species, but they can also be influenced by stress, as suspected in feline studies [13,44,53]. Therefore, stress may have falsely elevated LAC in stressed calves. Heart rate can also be increased by pain, especially in painful diseases like obstructive digestive diseases that involve mesenteric traction [5,41,44,52,54]. The severity of pain is a parameter used to guide surgical decisions in horses, and studies have shown that short-term survival rates are poorest in horses showing the most severe pain [40,41,43,52]. However, the impact of pain on the survival rate of calves suffering from surgical digestive diseases has never been studied. In our study, a pain assessment was performed subjectively for pain relief implementation; however, this evaluation was not included in our study. 

Overall, most studies suggest that a combination of parameters to evaluate shock and outcome will have more accuracy than a single parameter. Although the admission blood lactate values displayed prognostic significance, consecutive LAC measurement and changes in lactate in response to therapy may provide greater insight into patient survival [17,21,22,24,44,45,48,49]. Unfortunately, our protocol focused on admission shock parameters, and no measurement of lactate clearance was performed.

## 6. Perspectives

Similar to humans and small animals, various diseases in calves can lead to shock. Multiple forms of shock can and do occur simultaneously, and if left untreated, will lead to death. The main therapy of circulatory shock is based on the rapid administration of intravenous fluids to restore an effective circulating volume and tissue perfusion [3,55]. In ruminant practice, intravenous fluid administration is performed but it may not always be necessary or given in sufficient quantity, possible due to a lack of practical tools for evaluating shock in calves. Our study contributes to defining a combination of clinical signs and biochemical parameters that can be utilized for the early assessment of shock in calves. This early evaluation is crucial for initiating shock therapy before any other treatments are applied in calves that require it, monitoring the therapy, and thus improving the outcome of the disease. 

## 7. Conclusions

In conclusion, our study demonstrates that the initial blood L-lactate value is correlated with the heart rate, shock index, and clinical shock score in calves, and admission hyperlactatemia (LAC > 2 mmol/L) is related to a poor prognosis in calves under 4 months of age. Within this cohort, the shock index, defined as the HR/SBP ratio, in calves suffering from obstructive and non-obstructive digestive diseases was associated with mortality. The evaluation of the shock index could assist clinicians in identifying critical patients and improving outcomes through better shock management.

## Figures and Tables

**Figure 1 vetsci-11-00045-f001:**
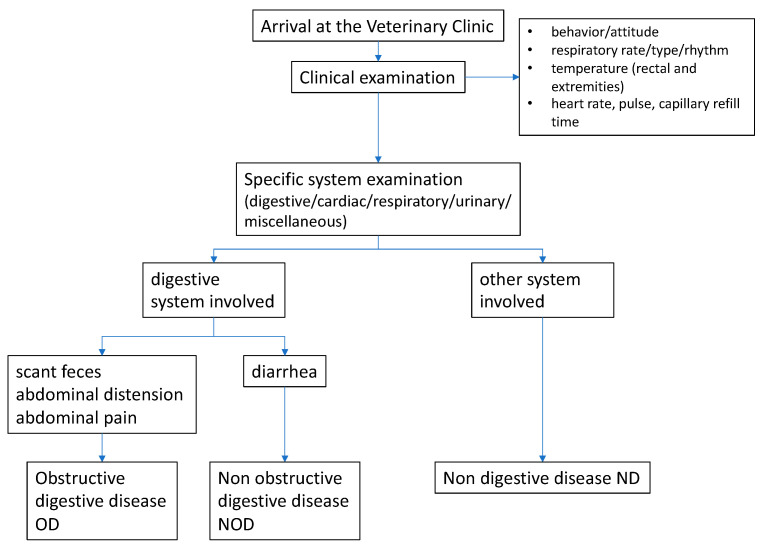
Flowchart of admission categorization of the calves involved in the study.

**Figure 2 vetsci-11-00045-f002:**
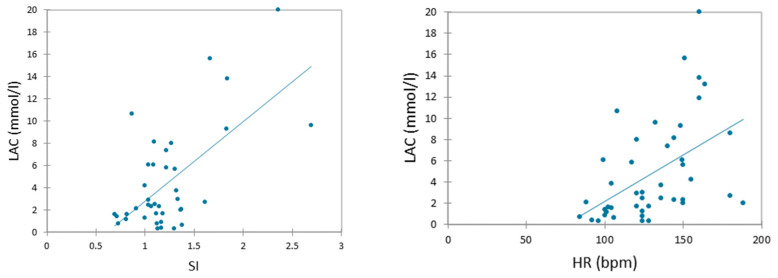
Scatter plots of Spearman correlation between blood lactate concentration (LAC) and heart rate (HR), and LAC and shock index (SI).

**Table 1 vetsci-11-00045-t001:** Clinical and biochemical continuous parameters of all calves involved in the study reported as mean and standard deviation (SD) or median and interquartile range (IQR) and *p*-values of Kruskal–Wallis bilateral test for CLINICAL SHOCK and DISEASE groups.

		n	HR (bpm)	CRT (s)	SBP (mmHg)	DBP (mmHg)	MBP (mmHg)	LAC (mmol/L)	SI
CLINICAL SHOCK	CSS	28	NA	NA	111 ± 23	63 ± 16	79 ± 17	3.73 (1.85–8.94) ^a^	1.20 (1.02–1.37)
	WOCSS	17	NA	NA	107 ± 25	59 ± 21	75 ± 22	2.10 (0.88–3.88) ^b^	1.12 (1.05–1.19)
DISEASE	OD	22	141 ± 22 ^c^	3 (2–3)	111 ± 20	61 ± 15	77 ± 16	4.20 (2.46–10.65) ^c^	1.15 (1.10–1.53)
	NOD	11	121 ± 30 ^d^	3 (3–4)	105 ± 24	60 ± 16	75 ± 18	2.05 (0.69–6.08) ^d^	1.16 (1.09–1.32)
	ND	12	119 ± 22 ^d^	3 (2–4)	112 ± 29	64 ± 25	80 ± 25	2.05 (1.38–2.45) ^d^	1.05 (0.88–1.24)
Total		45	130 ± 26	3 (2–4)	110 ± 23	61 ± 18	77 ± 19	2.60 (1.54–7.53)	1.14 (1.03–1.32)

n: number of calves; HR: heart rate; SBP, DPB, and MBP, respectively: systolic, diastolic, and median blood pressure; LACT: admission blood L-lactate; SI: shock index calculated dividing HR by SBP; CSS: clinical shock suspected; WOCSS: without clinical shock suspected; OD: obstructive digestive disease; NOD: non-obstructive digestive disease; ND: non-digestive disease; NA: not applicable; the values with different superscript letters in a column for each group “CLINICAL SHOCK” (a, b) and “DISEASE” (c, d) are significantly different (*p* < 0.05).

**Table 2 vetsci-11-00045-t002:** Clinical and biochemical continuous parameters of all calves involved in the study reported as mean and standard deviation (SD) or median and interquartile range (IQR) and *p*-values of Kruskal–Wallis bilateral test for the outcomes.

		N	HR (bpm)	CRT (s)	SBP (mmHg)	DBP (mmHg)	MBP (mmHg)	LAC (mmol/L)	SI
OUTCOME	P	21	124 ± 27	3 (2–4)	112 ± 24	63 ± 15	79 ± 17	2.0 (0.7–6.1) ^a^	1.1 (1.0–1.2)
	N	24	136 ± 25	3 (3–4)	108 ± 23	61 ± 21	76 ± 21	3.0 (2.1–9.5) ^b^	1.2 (1.1–1.5)
Total		45	130 ± 26	3 (2–4)	110 ± 23	61 ± 18	77 ± 19	2.6 (1.5–7.3)	1.1 (1.0–1.3)

N: number of calves; HR: heart rate; SBP, DPB, and MBP, respectively: systolic, diastolic, and median blood pressure; LACT: admission blood L-lactate; SI: shock index calculated dividing HR by SBP; OUTCOME positive (P) when the calf was discharged alive or negative (N) when the calf died or was euthanized during hospitalization; the values with different superscript letters in a column (a, b) are significantly different (*p* < 0.05).

**Table 3 vetsci-11-00045-t003:** The proportion of calves in each group (presence of hypotension (SBP ≤ 90 mmHg) or not (SBP > 90 mmHg), hyperlactatemia with LAC > 2 mmol/L or absence of hyperlactatemia, and hypotension combined with hyperlactatemia or absence of hyperlactatemia and hypotension), and their association with outcome.

		SBP ≤ 90	SBP > 90	LAC > 2	LAC ≤ 2	LAC > 2; SBP ≤ 90	LAC ≤ 2; SBP > 90
n		8	32	28	16	4	35
OUTCOME	P	4	13	10 ^a^	11	0	17
	N	4	19	18 ^b^	5	4	18

n: number of calves; SBP ≤ 90 and SBP > 90 in mmHg; LAC in mmol/L; OUTCOME positive (P) when the calf was discharged alive or negative (N) when the calf died or was euthanized during hospitalization; the values with different superscript letters in a column (a, b) are significantly different (*p* < 0.05).

## Data Availability

Data are contained within the article.

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
