# Peer review of "Evaluation of Blood Lactate, Heart Rate, Blood Pressure, and Shock Index, and Their Association with Prognosis in Calves"

_vetsci, 2024, doi:10.3390/vetsci11010045_

Round 1

Reviewer 1 Report

Comments and Suggestions for Authors

REVIEW OF THE PAPER vetsci-2745854-peer-review-v1

“Shock parameters evaluation and prognostic value in calves under 4 months of age”

The authors present a study to assess the prognostic value of shock parameters of sick calves up to four months of age and whether they are related to the outcome at the end of the disease. The work is interesting, it has practical interest in the calf clinic, especially in the veterinary hospital, which is where it is applied. However, before being accepted for publication, authors need to improve the manuscript.

- Table 2. I suggest that the “Total” value be put in the last row of the table.

A superscript (1,2,3; a,b,c) should be placed on statistically significant differences from the Kruskal-Wallis analysis between the OD-NOD, OD-ND, NOD-ND groups. I think it would be better understood by indicating the statistical differences between the groups in the table and then figure 1 would not be necessary.

- Line 205-207: I suggest a figure of the Spearman correlation between lactate and HR and lactate and SI. Although the text indicates that the correlations are good and significant, the image gives a better idea of what the correlation is like.

- Line 232-233: Do not give the sensitivity and specificity results as a percentage, but as an index has done in lines 274-279. However, the result can be offered in both ways, but it should be the same in all the manuscript.

- Table 3: I suggest that the “Total” value be put in the last row of the table.

- Lines 272-279: It is necessary to enter the 95% CI of the AUC because they are sample estimator values of population parameters and have an associated estimation error. Therefore, for any test to be statistically significant, the lower 95% CI value of the AUC must be > 0.5 (“Receiver operating characteristic curve: overview and practical use for clinicians”. Doi 10.4097/kja.21209).

- Lines 303-305: Author's quote at the end of the sentence.

- Line 328: Put the citation number in brackets.

- Line 340: Exchange for SBP

- Lines 339-352: The authors repeat the results, but do not give an interpretation of them. It would be interesting if these results were compared with those obtained in horses or small animals or in human medicine.

- Lines: 354-383: Limitations of study, it is correct and understandable, but it seems excessively long so it could be shortened.

Reviewer 2 Report

Comments and Suggestions for Authors

CONTEXT

The paper, titled “Shock parameters evaluation and prognostic value in calves under 4 months of age“ addresses an important and timely topic. I found the subject matter of the article fascinating, and read the manuscript with great interest. The paper aligns well with the scope of the journal, addressing a specific gap in the field. Compared with other published material, it adds, for the first time in the field of bovine clinical research, the evaluation of the shock index (SI) parameter, documenting the correlation between high admission blood L-lactates (LAC) with heart rate (HR) and shock index measurements in calves. However, I believe that in its current form it has some aspects that can be improved.

BRIEF SUMMARY OF THE PAPER

The single-observer prospective study, conducted on 45 calves aged 1 day to 4 months presented at the Veterinary Clinic for Ruminants in Liege from December 2020 to May 2022, had two objectives: 1) to evaluate the correlation between clinical hemodynamic parameters and blood lactates (LAC), systolic blood pressure (SBP), and shock index (SI) recorded on arrival and 2) to assess their correlation with short-term outcome in calves younger than 4 months of age presented in emergency. Statistical analysis showed a significant correlation between LAC and HR and LAC and SI. Further, a high LAC value on admission was significantly associated with a negative outcome (death).

Despite the strengths, there are some important limitations to consider, as already reported by the authors: 1) the failure to consider clinical parameters such as mental status and extremity temperature for the assessment of suspected shock state during the clinical examination, 2) the fact that HR and LAC may be affected by parameters such as stress and fear, which have not been assessed, and 3) the measurement of lactates on admission, but without subsequent monitoring.

SIMPLE SUMMARY

The simple summary is comprehensive and written in a way that non-experts in the field can understand the context and importance of the research conducted.

As a sole note, it is reported in this section that the study was conducted on 48 calves, but 45 calves are cited in the Abstract and in the Demographic Data of the Results. I would suggest correcting this numerical figure.

ABSTRACT

The abstract correlates with the manuscript content, including all the results and the significance of the obtained data.

KEYWORDS

To enhance the research's appeal, I suggest avoiding the inclusion of terms in the keywords that are already present in the article title.

INTRODUCTION

The introduction is clear and purposefully outlines the context of the study, referring consistently to recent literature.

MATERIALS AND METHODS

I suggest expanding the Methods section to provide a more detailed and comprehensive description of the procedures. This will enhance the clarity and replicability of your study. Consider including the following details: expand the method used to categorize calves according to their clinical condition (section 2.1. Animal and clinical evaluation) and specify the complementary tests performed and therapies given (section 2.4. Therapy and outcome).

RESULTS

The section is complete; it clearly lays out the data obtained.

DISCUSSION

I kindly suggest expanding the discussion section of your paper to include practical applications of the study. This addition will enhance the overall value of your research and provide a more comprehensive understanding of its implications.

CONCLUSION

The conclusions are consistent with the evidence and arguments presented and they address the main question posed.

REFERENCES

All the references are appropriate and included in the main text.

EDITING

There are some editing issues. It's recommended to thoroughly review the document for such problems.

Specific comments:

I recommend consulting another related paper that addresses the issue in different species using a similar methodology. Compare the methods employed and the results obtained. Please refer to the following: 10.3390/ani13061107.

I propose commencing the introduction with a depiction of the current management situation for modern dairy calves to provide the reader with a comprehensive overview. For further details, please refer to: 10.3390/vetsci10090554.

Enhance the statistical analysis by providing a more detailed methodology, accompanied by citations of appropriate references related to the chosen statistical methods. It is crucial to verify that the assumptions underlying your selected statistical methods are met. I suggest consulting the guidelines outlined in [proposed reference, e.g., 10.1186/s12917-022-03289-2 and 10.3390/ani12141740] for conducting these tests to uphold the rigor and reliability of your analysis.

Reviewer 3 Report

Comments and Suggestions for Authors

Shock in calves

I was interested in the concept of evaluating the outcome of treating young calves presented to a veterinary clinic/hospital for treatment of intestinal ‘catastrophes’ and thought this paper would address factors that positively or negatively influenced the outcome. After reading this paper, I was none the wiser! Why? Because of a lack of focus of the entire research methodology.

  1. what was the outcome variable? Was it survival following treatment, presentation of a defined clinical entity before treatment, estimate of shock index for each case? Nowhere in this paper was an objective of this research defined simply.
  2. without an outcome defined, it is impossible to investigate the  independent explanatory variables influencing the outcome. The paper is entitled ‘shock parameters’, but shock index has been well defined elsewhere - the ratio of heart rate over systolic blood pressure, an assessment of the vascular system following trauma of various origins. If they authors wanted to investigate other clinical/physiological/biochemical changes presented by calves admitted to their hospital with their shock index, fine: but that is never done here because the reader is never given the data.
  3. surely the outcome of a treatment depends on the standard of the initial diagnosis and clinical evaluation of the calf and the therapy administered? This paper infers that treatment outcome depends on the shock index prior to treatment; surely, the therapeutic strategy administered should be related to accurate clinical evaluation beforehand.  
  4. I have grave concerns regarding the case definition for inclusion within this study. Perinatal diarrhoea and endotoxaemia with associated metabolic acidosis is a well-defined pathophysiological clinical entity, associated with loss of bicarbonate via the intestine and increase in blood lactate and H+ ions in the peripheral blood circulation.Poor treatment outcomes originate usually from a veterinary clinician failing to recognise the physiological significance of the signs presented: in these cases, other clinical signs such as stance, recumbency, dehydration, heart rate give clear signals as to the therapy that must be provided, and a shock index is almost irrelevant - the vascular circulation in theses cases is compromised. However, for the abdominal ‘catastrophies’ that you define, clinical evaluation is much, much harder and to assess the degree of shock in a calf it is experiencing prior to surgery is vital. These two major clinical syndromes defined in this paragraph are totally different as far as therapy and outcome are concerned.
  5. this leaves you with a huge problem regarding numbers of cases included in this study: 22 cases of intestinal obstruction are too few for any statistical analysis of treatment outcomes; the 11 cases of neonatal scour are totally different in their aetio-pathology and the 12r calves suffering from diseases of other origins are unique in themselves. I fail to understand how the authors can include perinatal claves suffering respiratory distress syndrome in this study: shock does not enter into the clinical evaluation since the aetiology of the condition is aspiration of meconium/amniotic fluid during parturition and is a physical obstruction within the large bronchioles/bronchii and associated cellular inflammatory response (mostly of lymphocytes/mononuclear cells).
  6. the statistical analysis used in this paper is poor. First, there are only four outcomes when investigating the probabilities that different data sets are, indeed, different - n/s (not significant; 0.05 (significant at a 5% probability), 0.01 (significant at a 1% probability) and 0.001 (highly significant difference) Any other numbers given are irrelevant. The calves in the study are individuals - you cannot have 0.7 of a calf!! Elsewhere, I do not see how a shock index mean can be 1.1 and the SD 1.1 - 1.2 - it does not make sense.  Further, if you defined the outcome variable, you could carry out multiple regression analysis to investigate the factors contributing to that outcome - including surgical interventions with support for the vascular circulation. The simpler the statistical methods used, the better and more secure are the analytical results. 
  7. Most of the References quoted have little relevance to calf diseases: 24/47 are associated with equine/dogs/cat surgical management. I have no issue in principle with using References from human medicine - definition of shock index, its measurement, use in paediatric medicine -  but equine colic surgery and small animal abdominal surgery are totally different: the surgeon under those conditions is completely in the dark unless he/she knows precisely the pathophysiological aspects of vascular/circulatory dysfunction in the animal he is about to operate on. 

Title of paper: does not describe the subject of this paper.

Summary/abstract: what does ‘..field of young ruminant breeding…’ really mean? What generalised infections in calves under 16 weeks of age need resuscitation? Why do you need to evaluate shock, when shock index is clearly defined in medicine? For at least 11 calves, it is endotoxic shock you are dealing with, not septic shock. The statistical analysis has only one reasonable association, that of high peripheral blood lactate concentrations with heart rate: the remaining associations are all not significant. Calf mortality was not associated solely with  high peripheral blood lactate concentrations: other factors played a part. The numbers of calves in this study does not allow you to come to the conclusions you have made. Note: I define precisely what I am measuring; not high/low values, that could be anything!

Introduction; start at the beginning. The paper is about shock, so do not confuse the reader by digressing: stick to the topic. What is it and define it; how does it occur from a physiological/biochemical standpoint; types of shock; the likely clinical scenarios leading to shock. Specify the conditions causing shock in cattle - which is not the same as endotoxaemia/metabolic acidosis in calves: how best to measure shock index in cattle, if relevant. Respiratory acidosis is not the same as shock, either: In ll89 - 93, you confuse the reader by not defining your outcome and trying to find a relationship between variables that all affect the outcome of a therapeutic strategy and ignoring the treatment itself.

Materials and Methods: a case definition is absent, so that three or more very different clinical presentations are all lumped together. I suspect you have done this because of the small numbers of cases you encountered. There are papers in human medicine that evaluate clinical shock with other signs: why is this not in the Introduction? Section 2.4 is so general that it tells the reader nothing and suggests you have no therapeutic data recording outcomes. The numbers are so small that you have had to resort to non-parametric tests, and chi-square test should not be used on sample sizes less than 50: you never define the outcome variable you are investigating.

Results: You cannot ‘suspect’ shock was present: it either was or was not present. You cannot have 0.8, 0.9 of a calf, giving a mean for four cases - not clever! Table 1 makes no sense: the headings should be in full, not initials. The percentage totals are meaningless for small numbers - the reader can assess that for him/herself! There is nothing here that has not been described better in papers on shock in human medicine, and the numbers do no deserve a Table devoted to them. Table 2 is constructed badly: I have no idea what the outcome variable is, so the factors contributing to it are could be anything. All probabilities are not significant save for peripheral blood lactate concentrations between 28 calves with shock and 17 without at p<0.05, and similar probability for the same parameter in 22 calves with obstructive intestinal disease and 23 other calves. In ll 205 - 207, the values should be < 0.001 and <0.05 respectively, not as you have described them.Fig 1 tells us nothing that could not be covered in the text: I am sure that other clinical papers regarding treatment of abdominal catastrophes in cattle show the huge variation in heart rates and peripheral blood acidosis that you have recorded here. In Table 3, at last you have defined an outcome: discharged following treatment, or died: according to this data, only blood lactate concentrations before treatment were significant at the 5% level, and shock index - the subject of this paper - was not significant. Table 4 adds nothing to the data analysis. This is why this paper is so confusing to read or to understand the subject it is describing. 

Discussion: This does not appraise critically the data you present. The only relationships you have found relate to peripheral blood lactate concentrations, which is not the same as shock index - the title of this paper. Because the case definitions were never clear and a range of various clinical syndromes merged into one age-related group, the conclusions are insecure.

References: mostly unrelated to gastro-intestinal bovine medicine and surgery. The evaluation of many of the independent variables identified in this paper has made successful surgical treatments for equine colic now commonplace, but the same cannot be said for treatments for a range of clinical syndromes found in calves less than 16 weeks old. 

Comments on the Quality of English Language

Repetitive and not precise use of language.

Round 2

Reviewer 2 Report

Comments and Suggestions for Authors

The authors have diligently addressed the review comments, significantly enhancing the paper's quality. As a result, it is now well-suited for publication.

Author Response

We thank you for your thoughtful revision of the manuscript.

Reviewer 3 Report

Comments and Suggestions for Authors

A significant improvement with the rewrite of this paper, but a few issues remain which should be addressed easily. a) Simple Abstract - repeat five times of the word 'higher': alter style of English b) the English tense should be in the past throughout: reread and correct this c) the authors should consult references in human/veterinary reproductive physiology  regarding respiratory distress syndrome in neonatal calves - it is not a deficiency in surfactant but prolonged hypoxia through first/second stage labour. The basic pathophysiology was described way back in the twentieth century, first in lambs and applied to human obstetrics. 

Comments on the Quality of English Language

See above. Some words need removing that add nothing e.g. at start of Discussion, and some repetition remains. The Abstract contains phrases from the Introduction that seem 'cut and paste' - and please edit the Introduction so that each concept has its own identity with no repeat of References e.g. Ref 2.
